# Cadherin Signaling in Cancer and Autoimmune Diseases

**DOI:** 10.3390/ijms222413358

**Published:** 2021-12-12

**Authors:** Margherita Sisto, Domenico Ribatti, Sabrina Lisi

**Affiliations:** Department of Basic Medical Sciences, Neurosciences and Sensory Organs (SMBNOS), Section of Human Anatomy and Histology, University of Bari “Aldo Moro”, I-70124 Bari, Italy; domenico.ribatti@uniba.it (D.R.); sabrina.lisi@uniba.it (S.L.)

**Keywords:** cadherins, EMT, Sjögren’s syndrome, cancer, autoimmunity

## Abstract

Cadherins mediate cell–cell adhesion through a dynamic process that is strongly dependent on the cellular context and signaling. Cadherin regulation reflects the interplay between fundamental cellular processes, including morphogenesis, proliferation, programmed cell death, surface organization of receptors, cytoskeletal organization, and cell trafficking. The variety of molecular mechanisms and cellular functions regulated by cadherins suggests that we have only scratched the surface in terms of clarifying the functions mediated by these versatile proteins. Altered cadherins expression is closely connected with tumorigenesis, epithelial–mesenchymal transition (EMT)-dependent fibrosis, and autoimmunity. We review the current understanding of how cadherins contribute to human health and disease, considering the mechanisms of cadherin involvement in diseases progression, as well as the clinical significance of cadherins as therapeutic targets.

## 1. Introduction

Cadherin adhesion molecules, notably those of the classic cadherin subfamily, are critical determinants of tissue organization in both physiological and pathological conditions. Cadherins’ expression occurs through a dynamic process and is regulated by a great number of developmental and cellular signals. Since their discovery, it has become clear that the role of cadherins goes beyond simply promoting mechanical adhesion between cells. In fact, their function extends to multiple aspects of morphogenesis, ranging from the polarization of simple epithelia to the formation of tissues and organs architecture, conferring resistance to cellular detachment, controlling the morphogenesis of contacts as cells integrate into populations, and influencing tissue patterning and cohesion [1]. Given their highly variable functions, it is not surprising that abnormal or altered cadherins’ expression has also been linked directly to a wide variety of diseases including metastatic cancer and autoimmunity [2,3]. In this review we focus on the findings on cadherin-mediated cell adhesion systems in cancer and autoimmunity, highlighting the fundamental role of cadherins in epithelial-to-mesenchymal transition (EMT)-related diseases.

## 2. Cadherins

### 2.1. Types of Cadherins

The building and maintenance of a multicellular organism are based on the ability of cells to communicate and reciprocally adhere. By sensing their microenvironment, cells can migrate, proliferate, differentiate, change shape, move to a neighboring site, or die. The research studies in this field demonstrate that adhesion proteins activate multiple major signaling networks [1]. In the intercellular adherens junction, a core cadherin-catenin complex is present, that contributes to tissue stability and dynamic cell movements inside the tissue. Cadherins are transmembrane adhesion molecules that, based on sequence similarity, have been divided into five subfamilies: classical types I and II (E-, P-, N-, and VE-cadherin), atypical (T-cadherin), desmosomal (desmogleins, desmocollins), protocadherins, and cadherin-related proteins [4,5]. The classic cadherins family includes several types of proteins such as E (epithelial)-, N (neural)-, P (placental)-, VE (vascular-endothelial)-, R (retinal)-, and K (kidney)-cadherins; among these, E-cadherin is most frequently recognized in the formation of adherens junctions in epithelial cells. Classic cadherins found in the adherens junction are Ca^2+^-dependent cell-cell adhesion molecules; this structure presents an extracellular part consisting of five extracellular cadherin domains (ECs). Classic cadherin presents, in addition, a transmembrane domain and a carboxy-terminal cytoplasmic tail which binds to several adaptor molecules to transmit physical and biochemical signals to the cell [6]. The extracellular domain is responsible for homophilic interactions between cadherin molecules expressed at the surface of neighboring cells [6]. Cadherin cytoplasmic tails, on the other hand, bind to proteins p120-catenin and β-catenin (or alternatively, its homolog γ-catenin in some cell types) (Figure 1). While p120-catenin regulates the stability of cadherin-catenin complexes at the plasma membrane [7], β-catenin interacts with the actin-binding protein α-catenin, which contains an actin-binding domain and physically links adherens junction complexes to the actin cytoskeleton [7]. The integrity of the cadherin-catenin complex and its association with the cytoskeletal actin are prerequisites for cell-cell adhesion [7]. Interaction between the cadherin-catenin complex and the actomyosin cytoskeleton is regulated by mechanical forces and the Rho-family of small signaling GTPases [8]. These interactions, in fact, facilitate not only the coupling but also the detachment of cadherin-catenin complexes from cytoskeletal actin, allowing cell-cell separation, sorting, and migration.

### 2.2. Cadherin Trafficking Pathways

To modulate cell to cell adhesion, the regulation of cadherins’ expression on the cell membrane is a key point. In general, cadherin plasma membrane protein is in a defined equilibrium with intracellular endosomal compartments and the amount of this protein at the plasma membrane is determined by the balance of its endocytosis and its recycling. In fact, the appropriate steady-state levels of cadherin at the plasma membrane are determined by endocytosis and degradation trafficking, which determine a surface level decrease, and then the synthesis of new protein and recycling, which increase the amount of cadherin available at the plasma membrane [9]. The cadherin internalization and recycling pathway play a major role in the regulation of the protein surface level. Most scientific knowledge focused on this mechanism has derived from the clathrin-dependent pathway. In the last years, however, it has become clear that cadherin internalization occurs also via distinct clathrin-independent endocytosis pathways (Figure 2). Among these endocytosis mechanisms, clathrin-mediated pathways have been best characterized. Clathrin-mediated endocytosis of E-, N- and VE-cadherin occurs through the binding of clathrin adaptor proteins [10] Interaction with adaptor proteins is required for clustering the targeted receptors into clathrin-coated pits [11]. In accordance with the canonical Clathrin-mediated endocytotic pathway, the clathrin-coated pits invaginate into the cytoplasm and eventually bud off from the plasma membrane to form clathrin-coated vesicles [12]. The large GTPase, dynamin, is required to facilitate the fission process; this role is also essential in several other internalization pathways [13]. After endocytosis, internalized molecules can be recycled back to the plasma membrane, delivered to the lysosomes for degradation, or in polarized cells, sent across the cell through a process called transcytosis [14]. Clathrin-independent endocytosis is not well characterized; researchers have suggested that cadherin endocytosis may occur through both caveolin-mediated and macropinocytosis-like pathways. Akhtar et al. found that disruption of cell-cell adhesion in keratinocytes occurs through a dominant-active form of the small GTPase, Rac1, and this phenomenon was associated with E-cadherin caveolin-mediated endocytosis [15]. Lu and colleagues corroborated the evidence, demonstrating that E-cadherin internalization undergoes caveolin-mediated endocytosis in response to epithelial growth factor (EGF) in a human tumor cell line overexpressing EGF receptor (EGFR) [16], a mechanism which may be relevant to the EMT in cancers [16]. In contrast, Bryant and colleagues reported that EGF-dependent E-cadherin co-internalization occurs in a breast carcinoma cell line, along with cadherin-binding proteins p120 and β-catenin; EGF stimulation resulted in Rac1-modulated micropinocytosis, rather than caveolin-mediated endocytosis [17] (Figure 2). It is not clear if the EGF-related mechanisms described by Lu and Bryant are effectively different. Interestingly, Cadherin-6B, expressed in chicken pre-migratory cranial neural crest cells, has been reported to undergo both clathrin-dependent endocytosis and micropinocytosis [18]. A dynamin-dependent E-cadherin endocytosis mechanism, that was independent of clathrin- and caveolin-mediated macropinocytosis, was also described [19]. Finally, desmosomal cadherins undergo internalization through a lipid raft-mediated endocytosis, but it is currently unclear if this pathway is common also to classic cadherins [20]. While some of the specific details of the clathrin-independent pathways remain obscure, it appears that various cadherins can be internalized through different endocytic pathways under different circumstances, through both clathrin-dependent and clathrin-independent endocytic pathways. It is important to determine how the selected endocytic mechanism impacts cadherin function and how the contexts in which different trafficking decisions occur can modulate the cell fate and behavior.

### 2.3. Physiological Role of Epithelial and Vascular-Endothelial Cadherins

Adherens junctions and desmosomes are two intercellular junctions that maintain the integrity of the epidermis, controlling the differentiation and proliferation of keratinocytes. Classic cadherins mediate specific adhesion at the intercellular adherens junction. They share high sequence homology in their ectodomain with desmosomal cadherins [21,22,23]. E-cadherin and placental (P)-cadherin are classic epidermal cadherins. As to E-cadherin, it has been reported to be widely distributed in all skin layers and skin appendages, and its role in keratinocytes-melanocytes adhesion and communication has been established. The expression of P-cadherin, indispensable for skin and eye function, is more diversified and dependent on the skin layer. It is abundant in the basal and lower suprabasal layers, where it was shown to be linked with the proliferative cell compartment of the epidermis [24]. Desmosomal cadherins comprise two separate subfamilies, the Desmocollins (DSC, −1, −2, −3) and Desmogleins (DSG, 1, −2, −3, −4), all encoded by separate genes [22]. The expression of different members of the desmosomal cadherins presents a tissue-dependent temporal and spatial transcription [22]. The expression of the “skin type” desmosomal cadherins, Dsc1 and Dsg1, was correlated with the morphological features of keratinocyte terminal differentiation in the epidermis; they are expressed throughout the epidermis but most prominently in the upper layers. Dsg1 and Dsc1 expression decrease closer to deeper layers, whereas Dsg3 and Dsc3 are in most cases discovered within the basal layer and show an inverse expression gradient, reducing steadily in the parts closer to the top layers [25]. Dsg1/Dsg3 and Dsc1/Dsc3 are restricted to certain specialized, mostly stratified squamous epithelia, whereas Dsg2 and Dsc2 are the most ubiquitously expressed isoforms in desmosome-bearing tissues [26]. Furthermore, in the human epidermis, Dsg3 expression is detected from the basal to the spinous layers, whereas Dsg2 expression is limited to the basal cell layer [27]. Recently, a fourth member of the desmoglein family, Dsg4, was identified, which is expressed in the suprabasal layers of the epidermis [28]. VE-cadherin belongs to the classic type II cadherin type and is a strictly endothelial-specific adhesion molecule located at intercellular junctions (zonula adherens) between endothelial cells in all types of vessels, suggesting that it plays a role in cell-to-cell adhesion to maintain the normal architecture of the blood vessels [29]. Comparison of the amino acid sequence with classic type I cadherins, such as E-, N-, and P-cadherins, revealed only 23% identity for VE-cadherin [30]. VE-cadherin is the dominant adhesion molecule responsible for the maintenance and control of endothelial cell contacts; it is essential during morphogenesis of the blood vessel system, which is why VE-cadherin is one of the most intensely studied cadherins known today. Mechanisms that regulate VE-cadherin–mediated adhesion are important for the control of normal vasculogenesis and angiogenesis, for the maintenance of vascular integrity and permeability, and to regulate leukocyte extravasation in adults [29]. As is well known, stabilization of the VE-cadherin–β-catenin complex hinders vascular hyperpermeability and blocks leukocyte extravasation in inflamed tissues [31]. In addition to its adhesive functions, VE-cadherin regulates cell proliferation and apoptosis, thereby modulating vascular endothelial growth factor receptor functions. Consequently, VE-cadherin is essential during embryonic angiogenesis [32]. In fact, VE-cadherin is expressed in the embryo at very early stages, in mesodermal cells of the yolk sac mesenchyme, while at later embryonic stages, VE-cadherin expression is restricted to the peripheral layer of blood islands, giving rise to endothelial cells [33]. The role of VE-cadherin permeability control is consistent with the observation that the VE-cadherin-catenin complex is targeted by permeability-increasing agents, as reported in several recent studies. These demonstrated that mediators of inflammatory reactions such as thrombin, tumor necrosis factor-α (TNF-α), interferon-γ (INF-γ) and histamine, determine a decreased VE-cadherin expression during inflammatory tissue injury [34,35]. In addition, PMN adhesion directly affects cell-cell adhesion by inducing Src phosphorylation and SHP2 dephosphorylation of VE-cadherin at the binding sites for p120 and β-catenin. The adherens junction is then destabilized, dramatically altering the molecular composition and organization of VE-cadherin-catenin complexes in endothelial cells [36,37].

## 3. Changes in Cadherins’ Expression during the Epithelial- and Endothelial- to Mesenchymal Transition

Epithelial to mesenchymal transition (EMT) is a critical biological cell process that triggers profound morphological, molecular, and biochemical changes [38,39]. Hallmarks of the EMT are a series of events where epithelial cells are transcriptionally reprogrammed, resulting in a decreased adhesion, and the acquisition of a migratory capacity, alterations in cytoskeleton architecture, and other traits of a mesenchymal phenotype [38]. In several tissues, the cells that form epithelial layers present apical-basal polarity and are connected laterally by tight junctions and adherens junctions, the latter formed by E-cadherin molecules expressed on the cell surface [40,41]. This organization is fundamental for the structural integrity of epithelia. Epithelial cells which undergo the EMT show, on the contrary, the dissociation of cell-cell junctions and a reduced E-cadherin expression [41,42]. The epithelial cells lose their typical polygonal, cobblestone form and acquire a spindle-shaped mesenchymal morphology, expressing markers that are linked with the mesenchymal cell phenotype. In this context, the expression of E-cadherin and certain cytokeratins is drastically downregulated, while mesenchymal markers, prominently N-cadherin, vimentin, fibronectin, and β1/β3 integrins, are upregulated [39,43]. Nowadays, the concept of the EMT has shifted from being a binary process to that of a fluid phase in which cells exist along a spectrum of intermediate states [43]. During the EMT, cells present a plastic, transient and hybrid state called the partial EMT phenotype [44]. Cells in partial EMT have the characteristics of both epithelial and mesenchymal phenotypes [44,45]. These hybrid traits allow the cells to undergo collective cell invasion instead of the individual cell migration occurring in mesenchymal cells [46]. Various lines of evidence indicate that the EMT is orchestrated by several transcription factors (EMT-TFs), which act pleiotropically and in various combinations to promote the expression of genes that induce the mesenchymal cell phenotype, and to repress genes involved in the epithelial state [47]. The EMT-TFs that play the main roles in organizing EMT processes are the zinc-finger E-box-binding homeobox factors ZEB1 and ZEB2, Snail (also known as SNAI1), Slug (also known as SNAI2), and the basic helix–loop–helix factors TWIST1 and TWIST2 [39].

There is now considerable evidence that Snail represses the E-cadherin-encoding gene, *CDH1*, through binding to E boxes in the CDH1 promoter [48]. ZEB1 also represses *CDH1*, so inducing the expression of the genes that codify for vimentin and N-cadherin [49,50]. Furthermore, EMT-TFs also directly control the expression of several genes linked to cell polarity, such as *SNAIL* and *ZEB1*, regulating the expression of tight junction genes and components of apical-basal polarity [40,51]. Additionally, *Snail* and *ZEB2* trigger the expression of matrix metalloproteinases (MMPs), that provoke the degradation of the basement membrane, inducing cell invasion [40]. Recently, increasing reports have indicated several signaling pathways and messenger regulators that induce the EMT. These pathways have many common endpoints, including altered regulation of the expression of several cadherins, and the expression of EMT-associated genes that influence development, invasion, and metastasis in carcinomas, and organ fibrosis [41]. A key target of these transcriptional factors is the dramatic repression of the *E-cadherin* gene, an important keeper of the epithelial phenotype. The downregulation of the E-cadherin protein has multiple important consequences that are of direct relevance to the EMT [41,51]. The expression and activation of EMT-inducing transcription factors occur in response to several signaling pathways, including those mediated by transforming growth factor β (TGF-β), bone morphogenetic protein (BMP), epidermal growth factor (EGF), fibroblast growth factor (FGF), platelet-derived growth factor (PDGF), Wnt, Sonic Hedgehog (Shh), Notch, and integrin signaling [52]. These pathways signal through intracellular cascades, inducing the transcription of regulators that promote the expression of genes linked to the EMT process. Some of these signals may be more important in driving EMT at particular steps during the reorganization of the EMT process. Many EMT-inducing signals participate in crosstalk to integrate prompts from the microenvironment driving epithelial cell reprogramming [41,51]. TGF-β is the main signal-inducer of the EMT program in development and cancer progression; it controls various intracellular messengers such as Snail, ZEB, and the Twist family through Smad-dependent and Smad-independent signaling pathways [40,53]. Several lines of evidence highlight the key role of the Snail transcription factor in the EMT program switch [54,55]. In particular, it was demonstrated that TGF-β represses the expression of E-cadherin by regulating the transcription factor Snail. Indeed, the expression of Snail proteins was demonstrated to be inversely correlated with E-cadherin expression, and silencing of the *Snail1* gene can reestablish E-cadherin levels [56]. Furthermore, TGF-β signaling promotes the expression of ZEB proteins during the EMT process through an indirect mechanism, mediated in part by the transcription factor Ets-1 [40,56]. Successively, ZEB proteins interact with Smad3 and directly act by repressing *E-cadherin* gene expression, promoting cell migration [56]. Further findings have highlighted that also the Twist factor plays a key role, by downregulating E-cadherin and promoting the EMT through the TGF-β/Smad3 signaling pathway [57]. Interestingly, the over-expression of Twist occurs in a large number of breast cancers, with concomitant EMT activation and a consequently increased migration and invasion by the cancer cells [57]. In the last decade, emerging studies have explored the strategic role of the endothelial cells (ECs) in the tumor microenvironment [58]. Besides their key role in angiogenesis, ECs have been identified as capable of a remarkable level of plasticity, shown by their ability to change from an endothelial to a mesenchymal phenotype [59]. This phenotypic plasticity, named the endothelial-to-mesenchymal transition (EndMT), has been demonstrated in several pathological conditions [58,60] and in particular, in cancer, [61] being critically implicated in tumor metastasis progression [62,63]. The phenotypic transition of ECs to the mesenchymal state is associated with a notable decrease in endothelial markers such as VE-cadherin. Moreover, an altered expression of VE-cadherin was demonstrated in some cancer types. In particular, in breast carcinoma, VE-cadherin was shown to promote cancer invasion and metastasis through increasing TGF-β signaling [64]. Recently, many studies have suggested that increased TGF-β signaling is a common underlying mechanism in almost every EndMT-linked disorder [65]. TGF-β family members mediate the EndMT via Smad or non-Smad signaling, promoted by inducing the expression of specific transcription factors, such as Snail, Slug, Twist, ZEB [66]. TGF-β interacts with other signaling pathways that mediate and/or regulate the EndMT, such as the Notch [67,68], fibroblast growth factor (FGF), Wnt, and Sonic Hedgehog pathways [69]. Interestingly, blocking TGF-β signaling might be a promising therapy for EndMT-related diseases and EndMT is a promising target for cancer therapy, although more investigation is needed in this field.

## 4. Cadherin-Mediated Signaling in the Context of Disease

Cadherin-mediated signaling mechanisms have been proven to play a crucial role in many malignancies and pathological states. One of the most well-studied examples is metastatic cancer. A wide variety of previous reviews explored cadherin-dependent regulation of tumor proliferation, also studying invasiveness, tumor cell metabolism, and metastasis. Although E-cadherin is a classic example of cadherin/EMT mediated signaling in cancer, N-cadherin, VE-cadherin, desmosomal cadherins, and other cadherins have also been shown to participate in oncogenic signaling in the pathogenesis of cancer. The following paragraphs seek to analyze the complex, prolific knowledge about cadherins’ role during pathological states, particularly focusing on cadherins’ signaling in the context of oncogenesis. In addition, an innovative field of investigation is examined, that probes the cadherin contribution to autoimmune diseases.

### 4.1. Cadherin and Tumorigenesis

#### 4.1.1. Interplay between Cadherin, Tumorigenesis and the EMT

It is nowadays well established that the EMT is associated with tumor onset, invasion, metastasis, and resistance to therapy. In carcinomas, the exchange of signals among cancer cells and their microenvironment is mostly responsible for EMT activation and hence the acquisition of motile mesenchymal phenotypes and malignant progression [70,71,72]. Initially, a plethora of stimuli can trigger the EMT; most of these events converge, inducing a downregulated expression of epithelial proteins, including those that are part of cell junction complexes [40,44]. Therefore, these processes undergo a significant change in the signaling pathways that define cell shape, reprogramming the gene expression to trigger reorganizational changes of their cytoskeletal architecture, to promote adhesion to mesenchymal cells, and to alter the interaction of cells with the extracellular matrix (ECM) [39,40]. The hallmark of the EMT is the deregulation of cadherins, which promotes the destabilization of adherens junctions and induces variations in EMT-associated gene expression profiles. The broad range of cadherins activities on structural tissue organization makes them attractive targets during tumorigenesis when their disruption can contribute to aberrant morphogenesis in cancer [73,74,75].

#### 4.1.2. E-cadherin and N-cadherin Deregulation and Cancer Development

In the past few years, having been recognized as diverse and multifactorial, cadherins’ expression has been widely studied in human cancers [76,77,78]. When a clearly differentiated benign adenoma with apical polarity becomes an invasive carcinoma, losing its normal polarization and forming metastases, cell-cell adhesion is strongly decreased [79]. Classic cadherins such as E-cadherin and N-cadherin are key regulators in the process of malignant tumor development [80]. Typically, N-cadherin is widely expressed in the nervous system and modulates the intercellular adhesion proteins of neurons, while it is expressed at low levels in other normal cells [81]. In recent years, it was demonstrated that N-cadherin endows tumor cells with an enhanced migratory and invasive capacity, resulting in the acquisition of an aggressive tumor phenotype. Thus, its defective overexpression is linked with tumorigenesis and metastasis [82]. The aberrant expression of N-cadherin in epithelial cancer cells is a well-documented feature of epithelial malignancies, and an abnormal expression of N-cadherin has also been found in many other tumors, such as lung cancer, hepatic cancer, urothelial cancer, and prostate cancer, and is associated with disease progression [83,84,85,86]. In addition, the soluble N-cadherin level in the serum of cancer patients is much higher than that in the serum of healthy patients, revealing a positive relation with poor prognosis [87]. Recently, it has been shown that N-cadherin allows the metastatic behavior of tumor cells by directly mediating cell-cell adhesion, and through its involvement in modulating critical signaling pathways such as TGF-β1, Wnt/β-catenin, EGFR, and NF-κB [82]. Moreover, N-cadherin is overexpressed in invasive and metastatic breast cancer, inducing metastasis by boosting FGF receptor signaling [87]. Interestingly, it was documented that knocking down N-cadherin inhibits the invasiveness of human melanoma cells [87]. In fact, further study in vitro has demonstrated that N-cadherin downregulation reduces the aggressiveness of esophageal squamous cell carcinomas [88]. The most frequently evaluated and extensively studied cadherin is E-cadherin [78]. E-cadherin is the core component of epithelial adherens junctions and is considered a tumor suppressor protein. The loss of E-cadherin expression is associated with the epithelial EMT and occurs frequently during tumor development. Indeed, the role of E-cadherin in carcinogenesis is of great interest, since it is an important determinant of tumor progression, serving as a suppressor of invasion and metastasis in many contexts. A recent finding by Li et al. in breast cancer patients showed that significant loss of E-cadherin expression is sufficient to confer a metastatic ability to breast cancer cells that are otherwise essentially non-metastatic [75,89]. Many factors such as mutations, proteolytic cleavage, chromosomal deletions, epigenetic regulation, and transcriptional silencing of the CDH1 promoter, were shown to impede the functionality of E-cadherin during the development of several malignancies including gastric, breast, liver, pancreas, and skin cancers [41]. Numerous missense mutations in the *E-cadherin* gene had been reported to interfere with the expression and function of this protein [90]. Of particular interest, one of these *E-cadherin* gene mutations has been implicated in the pathogenesis of diffuse gastric cancer, suggesting that genetic deficits in E-cadherin contribute to cancer development and progression [90,91]. Additionally, patients with inherited *E-cadherin* gene mutations have been reported to present a substantial risk for developing many epithelial malignancies, including esophageal and hepatocellular carcinoma and melanoma [92]. Besides, deletional mutational analysis in the *p120* gene, a regulator of E-cadherin, has demonstrated that decreased p120 expression causes the degradation of E-cadherin in lung cancer [93]. Interestingly, increasing studies have also been devoted to exploring the prognostic role of E-cadherin. Indeed, the loss of E-cadherin expression was considered to have a particular prognostic value in breast cancer, where it may even be more informative than tumor size or estrogen receptor expression [94]. In addition, a decreased expression of E-cadherin and β-catenin, as confirmed by immunohistochemistry, are important prognostic markers in patients with bladder carcinoma [95], and were linked with high grade and invasive stage of bladder carcinoma [96]. Many other studies have described a relationship between decreased E-cadherin and/or catenin expression, correlated with dedifferentiation, infiltrative tumor growth, distant metastasis, and poor survival for patients with gastric carcinoma [97], pancreatic carcinoma [98], prostate cancer [99]. Interestingly, recent findings have demonstrated that the term leakage of E-cadherin expression is an oversimplification because many metastases still contain high levels of E-cadherin, and epithelial cells expressing E-cadherin can become invasive and metastasize without undergoing a full EMT program in tumors [100]. Indeed, Kowalski et al. [101] hold the view that abnormal E-cadherin expression is more common in invasive ductal carcinomas with the potential to develop distant metastases, and the E-cadherin expression is more consistent and often more frequent in the distant metastases than in primary cancer. Interestingly, E-cadherin may be re-expressed in distant metastases of invasive cancer, suggesting that it may contribute to the formation of metastatic foci [101].

#### 4.1.3. Other Cadherins and Tumor Progression

Researchers have focused further study on other types of cadherins that have similar effects to E-cadherin in malignant tumor formation. For example, P-cadherin dysfunction is strongly linked with growing tumors, conferring the malignant phenotype to cancer cells [102,103]. Indeed, studies have demonstrated that P-cadherin acts as a tumor suppressor as its absence is associated with a more aggressive cancer cell phenotype [104]. The role of P-cadherin in human cancer is still debated and remains doubtful as it can behave differently depending on the cellular context and experimental cell model used for the study [105,106]. For example, in lung carcinoma, melanoma, oral squamous cell carcinoma, and hepatocarcinoma, P-cadherin has similar tumor-suppressive behavior to that of E-cadherin. Nevertheless, recently, overexpression of P-cadherin has been shown to be a feature of breast, ovarian, prostate, endometrial, skin, gastric, pancreas, and colon tumors, pointing to its induction of aggressive behavior [105,106]. More recently, many studies have been focused on another player, VE-cadherin, which takes part in an intricate interplay of classical cadherins in several cancer progression forms. VE-cadherin increases the ability of fibroblastoid cancer cells to proliferate, forming metastatic structures and adhering to endothelial cells, characteristics that are typical of aggressive behavior and malignant potential. Interestingly, aberrant VE-cadherin expression seems to be linked to certain cancer types; in breast carcinoma, for example, VE-cadherin was shown to promote tumor cell proliferation and metastatic invasion [107]. Analysis of the signaling cascade showed that VE-cadherin expression regulates Smad2 phosphorylation and the expression of target genes of the TGF-β pathway [76]. Several lines of evidence confirm that VE-cadherin might thus promote cancer progression and metastasis, not only by stimulating angiogenesis but also by increasing tumoral cell proliferation through TGF-β activity [64,107].

### 4.2. Autoantibodies against Cadherins as Markers for Autoimmune Diseases

Several antibodies blocking VE-cadherin have been described, which can interfere with the endothelial-specific adhesion properties of the protein [108,109,110]. Vascular endothelium and blood vessels are often implicated in the pathogenesis of chronic inflammatory autoimmune diseases including rheumatoid arthritis (RA), systemic lupus erythematosus (SLE), systemic sclerosis (SSc), and Behçet’s disease (BD). Based on this evidence, several recent reports have analyzed anti-VE-cadherin autoantibodies in sera from patients with autoimmune diseases such as RA, SLE, and BD but, interestingly, not in SSc. In addition, in Pemphigus, an autoimmune blistering disease, autoantibodies were found, directed against desmosomal cadherins and against E-cadherin and P-cadherin, classic cadherins expressed in the epidermis, identifying these cadherins as additional immunological targets in autoimmune conditions [111,112,113]. In the following paragraphs, we report the recent discoveries in this field, underlining the prognostic-predictive role of antibodies directed against cadherins in autoimmune diseases. A schematic representation of the mechanism of action of anti-VE-cadherin autoantibodies is reported in Figure 3A.

#### 4.2.1. Rheumatoid Arthritis

RA is a chronic inflammatory disease in which the main site of inflammation is the synovium. The presence of anti-VE-cadherin autoantibodies in RA is under debate. Rheumatoid vasculitis is a condition that features inflammation of blood vessels, that arises in some patients who have suffered from RA for a long time. The association with vasculitis alters the course and prognosis of the disease, being a significant cause of mortality [114]. Systemic vasculitis develops in some RA patients and not others; the reason for this is unknown but researchers have highlighted the fact that patients with associated vasculitis show soluble VE-cadherin, and that these levels are correlated with the disease activity score [115]. Probably, in patients with long-standing RA, the damaged endothelium determines VE-cadherin exposure to specific autoantibodies (Figure 3A). The underlying molecular mechanisms leading to anti-VE-cadherin autoantibodies production need to be clarified, comparing anti-VE-cadherin autoantibodies in patients with RA and vasculitis versus patients with RA without vasculitis. Moreover, further clinical-immunological studies are required to pinpoint the relative roles of the possible immunological therapies [116].

#### 4.2.2. Systemic Lupus Erythematosus

##### Role of Ve-cadherin as Marker for SLE

Vasculopathy is common also in SLE. Problems in the central nervous system (CNS), blood, skin, kidneys, gastrointestinal tract, and lungs could be observed in patients with vasculopathy associated with SLE [117]. These observations, obtained many years ago, led Ding and collaborators [118] to classify several pathological types of SLE vasculopathy: vascular immune complexes deposit, non-inflammatory necrotizing vasculopathy, thrombotic microangiopathy, and true lupus vasculitis. In these vasculopathies, antibodies directed against endothelial cells are detected, that exert cytolytic functions against vascular endothelial cells and have been associated with nephritis in SLE patients [119]. Anti-VE-cadherin autoantibodies have been shown in SLE [108]; in fact, since SLE can affect all organs and systems, critical manifestations are observed in kidneys, joints, skin, and CNS. This reflects the heterogeneity of anti-VE-cadherin autoantibodies levels determined in the different SLE patient groups, corresponding to their heterogeneous clinical status. The characterization of anti-VE-cadherin autoantibodies specificity for the target epitopes antigens showed that in SLE patients, anti-VE-cadherin autoantibodies preferentially recognized the EC1 fragment of the extracellular part of VE-cadherin [108]. The pathogenesis of these anti-VE-cadherin autoantibodies might be of clinical interest, although future studies will need to confirm and fully establish the clinical relevance of anti-VE-cadherin autoantibodies in SLE (Figure 3A). Interestingly, since the onset of CNS vasculitis is one of the major complications of SLE, it would be useful to evaluate whether the levels of anti-VE-cadherin autoantibodies can be indicators of an evolution of the autoimmune disease towards the form characterized by damage to the endothelium of blood vessels in the CNS. In clinical terms, if so, anti-VE-cadherin autoantibodies could be used as a useful marker for SLE [120].

##### Role of E-cadherin as Marker for SLE

In addition to the detection of anti-VE-cadherin autoantibodies, ligand to membrane-bound E-cadherin, αEβ7/CD103, was found on human peripheral blood lymphocytes derived from SLE patients. Furthermore, elevated αEβ7/CD103 expression was associated with oral ulcers or serositis in SLE patients [121]. The increased interaction of αEβ7/CD103 ligand with E-cadherin might contribute to epithelial inflammation, which is characteristic of SLE [121]. In addition, a situation of chronic inflammation, leading to elevated MMP through the stimulation of cytokines and inflammatory mediators [122], is known to cleave membrane-bound E-cadherin and release soluble E-cadherin (sE-cadherin) [123]. Serum levels of sE-cadherin could reflect the disease activity index of SLE. Indeed, the levels of sE-cadherin were positively correlated to s-creatinine, age, erythrocyte sedimentation rate, triglycerides, and elevated sE-cadherin levels detected in SLE patients with renal damage, suggesting a possible relationship between lipid metabolism and E-cadherin shedding in patients with SLE [120].

#### 4.2.3. Behçet’s Disease

In BD, a chronic multisystemic inflammatory disease, vasculitis affects large- and medium-sized arteries, and thrombosis is associated with vessel destruction and inflammation [124]. Currently, the diagnosis of BD relies on clinical manifestations, because no diagnostic test is available. Recent studies have indicated the detection of anti-VE-cadherin autoantibodies as a promising marker for the diagnosis of BD. The results obtained by Bouillet and collaborators demonstrate, in purified immunoglobulin G from BD patients, exceptionally high anti-VE-cadherin autoantibodies levels that recognized the extracellular module EC1-4 of VE-cadherin. More in-depth analysis indicates that the epitopes recognized by anti-VE-cadherin autoantibodies in BD patients were mainly restricted to module EC4 or the inter-module EC3-EC4 region. As it is well known that anti-VE-cadherin autoantibodies could disrupt the inter-endothelial adherens junction, they may have a pathophysiological role in the vascular lesions associated with BD disease [108]. Although the precise pathogenic role of autoantibodies directed against cadherins is still unknown, various hypotheses can be advanced. For classic cadherins (such as E-, P-, and N-cadherin), several studies indicate that monoclonal antibodies able to blockade cadherins activity produce this effect through binding to the cadherin extracellular domain EC1, which is responsible for homophilic recognition [125]. As regards the VE-cadherin activity, synthetic antibodies have been developed in the laboratory as tools to clarify the biological functions of VE-cadherin in vitro. It was clearly demonstrated that these antibodies can disrupt the adherens junction between cultured endothelial cells [126]. In addition, these synthetic anti-VE-cadherin antibodies were able to prevent the protective effect of VEGF on apoptosis [126]. These monoclonal antibodies probably act by inducing a conformational change in the VE-cadherin structure, which alters the structure of the amino-terminal region by preventing homophilic binding (Figure 3A). This hypothesis was supported by several studies demonstrating that conformational changes in other cell adhesion molecules, as a result of antibody/ligand binding, could activate or inhibit their adhesive properties [127,128]. Another interesting possible mechanism of action was proposed after an in-depth study of C-cadherin, which presents successive domains along the full extracellular segment EC1-5. The strongest adhesive interaction occurs between the fully interdigitated antiparallel proteins. Using antibodies to block even only one site on C-cadherin, the force of adhesion mediated by cadherin can be altered, thus facilitating AJ disassembly and rupture [129]. Based on this experimental evidence, it is possible to hypothesize that anti-VE-cadherin autoantibodies probably act in the same way as synthetic antibodies, by increasing vascular permeability and facilitating leukocyte recruitment and apoptosis, events that are often observed in autoimmune vascular lesions.

#### 4.2.4. Pemphigus

Pemphigus is a rare but severe blistering disease affecting the skin and mucous membranes, which is characterized by the presence of pathogenic autoantibodies targeting the desmosomal cadherins desmoglein (Dsg) 1 [prevalent in Pemphigus foliaceus (PF)] and Dsg3 [prevalent in Pemphigus vulgaris (PV)] [130]. In Pemphigus, like in the diseases reported above, these autoantibodies are pathogenic [131,132,133,134] and recognize epitopes located on the EC1–2 domains of Dsg [135,136,137,138]. Another aspect that has attracted great attention in the last years is the identification of E-cadherin as a possible additional immunological target for Pemphigus autoantibodies. Interestingly, anti-E-cadherin antibodies were detected only in Pemphigus patients with cutaneous disease, and this presence was correlated with the detection of anti-Dsg1 autoantibodies in Pemphigus patients. This observation suggests that a population of autoantibodies capable of recognizing epitopes present on both E-cadherin and Dsgs antigens probably exists, due to the great degree of homology between these molecules. Even if the potential pathogenic role of the anti-E-cadherin autoantibodies in Pemphigus is still unknown, several observations indicate a link between E-cadherin autoantibodies and Pemphigus acantholysis, with loss of cohesion between epidermal cells due to the breakdown of intercellular bridges [139,140]. In addition, an increased immunoreactivity of P-cadherin was also detected in the lesional skin in Pemphigus, suggesting that this upregulation is common to both the autoimmune bullous skin diseases which constitute inherited acantholytic diseases [141]. These autoantibodies determine p38/MAPK phosphorylation in pemphigus skin lesions [142], as confirmed by pharmacologic p38MAPK inhibition, that blocked blister formation in response to both PV–IgG and PF–IgG [142]. In a related study performed in Yasuo Kitajima’s laboratory, protein kinase C (PKC) was the first signaling molecule shown to be activated by Pemphigus autoantibodies [143,144,145]. Recently, the fact that inhibition of PKC abolished loss of keratinocyte adhesion in vitro and blister formation in vivo [146] was reported. Based on all these observations, a group of researchers investigated the role of signaling mechanisms in the regulation of desmosomal adhesion mediated by specific desmosomal cadherin. Available data suggested that modulation of extracellular binding of desmosomal cadherins triggers a signaling cascade of molecular events inside the cell [147]. This can be concluded because autoantibodies directly interfering with Dsg3 binding induces rapid alterations of signaling pathways. Furthermore, peptides designed to interfere with Dsg interaction [148] induced the activation of p38MAPK [149]. The way this extracellular cue, loss of binding, is transmitted to the cell is still unclear. Specific signaling molecules associated with the Dsg3 complex, for example, Rho GTPases and Src, seem to be involved [150]. In this complicated scenario, it should be remembered that PKC has been located at the desmosomal plaque [151], and may directly bind to desmosomal proteins, while p38MAPK is attached to a complex containing Dsg3 [149]. Furthermore, p38MAPK is activated when challenged with Dsg3-specific antibodies, and the activation of PKC by Pemphigus autoantibodies seems to be mediated by a rapid increase of cytosolic Ca^2+^ [152] stimulated by the phospholipase C/inositol 1,4,5-trisphosphate pathway, although the related activation mechanism of the latter is still unknown [153].

### 4.3. Cadherin-Dependent EMT in Autoimmune Diseases: Recent Advances

#### 4.3.1. Rheumatoid Arthritis

In RA progression, inflammatory synovium tissue undergoes severe remodeling characterized by fibroblast proliferation, and by the formation of an invasive hyperplastic tissue called the pannus [153]. Pannus tissue can invade the cartilage matrix, mediating the degradation of cartilage and bone [154]. Pannus tissue contains an elevated number of activated fibroblast-like-synoviocytes (FLS) and macrophages. In a healthy synovial membrane, FLS are essential to establish the complex synovial lining architecture, characterized by a multicellular organization and the production of synovial fluid [155]. In the inflamed rheumatoid synovium, the complex three-layer lining structure is converted into the hyperplastic synovial lining of a pannus-like structure, in which activated FLS and macrophages extend within the joint space and attach to the cartilage surface (cartilage–pannus junction), leading to cartilage matrix damage. These mechanisms are responsible for the joint destruction observed in RA [156]. As early as 2006, it was highlighted that the changes that occur in the synovial lining during the development of inflammatory RA resemble the changes that occur during the EMT process. In fact, the epithelial cells that form the peritoneal lining become hyperplastic and show a transformed mesenchymal phenotype (myofibroblast phenotype). By immunostaining, Steenvoorden and colleagues demonstrated a reduction in epithelial marker E-cadherin and an increased expression of α-smooth muscle actin (α-SMA) in the synovial tissue of RA patients as compared with healthy subjects. They proposed that under the stimulus of transforming growth factor-β (TGF-β), a key player of EMT activation, healthy *synovial fibroblasts* (SF) might undergo a process comparable to the EMT in RA patients synovial fluid [157]. These observations were confirmed by the finding that hypoxia-induced EMT was accompanied by increased migratory and invasive phenotypes in RA SF [158]. Results reported from these two studies raised controversy about the fact that the synovial lining lacks a basement membrane and should not express E-cadherin; in addition, SF are already mesenchymal cells and should not undergo the EMT or any other process resembling the EMT. However, various factors involved in the EMT process are found in rheumatoid joints: TGF-β is abundantly expressed in the synovial fluid of RA patients [159], while Slug was detected in the synovial tissue of RA patients and correlated with the invasive phenotype of RA SF [160]. Furthermore, a strong expression of α-SMA in the highly inflamed synovium of RA patients was detected, suggesting its presence on fibroblast-like synoviocytes in the synovial lining [157]. Although this evidence clearly indicates a potential role of the EMT in the rheumatoid synovium, further investigations are needed to solve these critical points and to identify the role of the EMT in rheumatoid joints. The hypothetical role of the EMT in RA is schematized in Figure 3, panel B. Recently, RA research was focused on *Cadherin-11 (CDH11)*, the gene located on chromosome 16q22.1. In joints, CDH11 is mainly expressed in FLS that regulates migration, invasion, and degradation of joint tissue, playing a significant role in the etiopathogenesis of RA [161]. Park and collaborators [162] demonstrated that CDH11 expression in FLS is regulated by IL-17 levels, which could aggravate synovitis and bone destruction. In RA, CDH11 leads to the aggregation of angiotensin cell clusters, promotes the invasion of angiotensin into the articular cartilage, allows the pannus to extend and pathologically invade the joint cartilage, and induces FLS to produce proinflammatory mediators enhancing the chronic inflammatory response in RA [163]. CDH11 seems therefore to be a promising therapeutic target in RA, because systemic administration of anti-CDH11 antibodies reverses the proliferation and migration of synoviocytes to the sites of joint inflammation, attenuating the symptoms of RA [164] The arthritis drug celecoxib has the structural potential to bind CDH11 and might function as CDH11 antibody-based therapy in clinical trials for RA.

#### 4.3.2. Recent Advances in Understanding of the Role of Cadherins in Sjögren’s Syndrome

##### Soluble E-cadherin in Sjögren’s Syndrome: Murine Models

Sjögren’s syndrome (SS) is a chronic inflammatory autoimmune epithelitis characterized by complex pathogenesis, that affects the exocrine glands, mainly the lachrymal and salivary glands (SGs), resulting in symptoms of oral and ocular dryness. SS is classified as primary (pSS) when the clinical manifestations occur alone, or as secondary form (SS) that complicates or overlaps with other rheumatic conditions. [165]. Based on the recent literature, the epithelium plays a pivotal role in orchestrating the focal lymphocytic infiltration of the exocrine glands which is the recurrent histological hallmark of pSS. These observations have been enriched by a recent investigation of the serum levels of soluble E-cadherin (sE-cadherin) to characterize the expression of E-cadherin and integrin alphaE beta7/CD103 (associated with infiltrating lymphocytes) in SGs epithelium of patients with pSS [166]. Interestingly, serum levels of sE-cadherin were significantly increased in pSS compared to non-SS controls and the elevated levels of E-cadherins did not seem to be associated with the formation of the germinal center. E-cadherin was detected on the majority of acinar and ductal epithelial cells surfaces in both pSS and non-SS patients. Focal infiltrates in the germinal center contained scattered alphaEbeta7/CD103-positive cells, which were detected also in small clusters near ductal and acinar epithelial cells. Interestingly, increased levels of these cells are present in pSS compared to healthy controls [166]. Double-labeling revealed that the E-cadherin-positive cells were CD68(+) macrophages. The elevated serum levels of sE-cadherin detected in pSS patients certainly indicate an increased epithelial cell turnover and shedding, as circulating sE-cadherin reflect tissue-specific regenerative processes or turnover. It remains to be clarified whether sE-cadherin can modulate the activity of neighboring cells or simply derive from MMP-cleavage. Actually, a correlation between serum sE-cadherin and proteolytic damage of ECM [167] and SGs destruction induced by MMP in pSS [168] is deemed possible. These hypotheses have been supported by the NOD mouse model of SS; in the NOD mouse, E-cadherin is ectopically expressed by mononuclear cells in the SGs, and T cells expressing E-cadherin in SGs show an altered proliferative response to immobilized anti-CD3 antibody [169]. Furthermore, the alphaEbeta7/CD103 molecule is present on the surface of CD8+ tissue-resident memory T (CD8+ TRM) cells that can bind E-cadherin on epithelial cells [170,171]. In murine SS, the infiltration of pathogenic CD103+CD8+TRM cells in glandular tissues could explain why the SGs are typically damaged [171].

##### E-cadherin in EMT-Dependent Fibrosis in Sjögren’s Syndrome

In the last years, more in-depth study of E-cadherin in EMT-dependent fibrosis in pSS has been made [172,173]. In the markedly inflammatory microenvironment of pSS SGs, the loss of E-cadherin from epithelial cells, and the acquisition by some cells of mesenchymal markers such as vimentin and collagen type I, was clearly demonstrated [172]. In addition, pro-inflammatory cytokines such as IL-6, IL-17, and IL-22 can trigger EMT-dependent SGs fibrosis. In the presence of fibrosis, SGs showed a decreased E-cadherin cellular expression [173,174]. This process occurs through the activation of both SMAD-mediated and SMAD-independent pathways, that lead, in a canonical or non-canonical manner, to activation of the IL-17 and IL-22-dependent EMT process in pSS [174]. The chronic inflammatory situation determines a decreased gene and protein expression of E-cadherin, accompanied by increased levels of vimentin and collagen type I, for example, owing to the pro-fibrotic activity of IL-6 [48,172,173,174]. The role of cadherins in EMT-dependent fibrosis in pSS is represented in Figure 3, panel B. The activation of EMT-dependent fibrosis in pSS could be explained by evaluating the expression of the factors involved in the cascade of the TGF-β1/EMT-dependent fibrosis pathway, which resulted in altered in pSS tissue compared with non-SS [172,173,174]. These observations were confirmed by the finding that healthy human salivary gland epithelial cells, when exposed to TGF-β1 stimulation, acquired a more fibroblast-like morphology, characteristic of the EMT (Figure 3B).

## 5. Concluding Comments

This review clearly demonstrates that we cannot yet provide any simple answer to the question of how cadherins dysfunction may promote progression in pathological conditions. The enormous variability of the processes in which cadherins seem to play a determining role implies that their dysregulation could lie at the basis of disease states encompassing epithelial barrier defects, inflammation, neoplasia, and autoimmunity. The emerging notion that abnormal activation of EMT contributes to tumor invasion and metastasis, and fibrosis during chronic inflammatory autoimmune diseases, is illustrative of the major physiological impact of cadherins, that by regulating morphogenesis, can become pathogenic when aberrantly expressed. While great pioneering research has been performed, we have only just begun to understand the extent to which cadherins participate in signaling events. Further studies are warranted and may support the development of novel strategies for both the prevention and treatment of cadherin-mediated diseases.

## Figures and Tables

**Figure 1 ijms-22-13358-f001:**
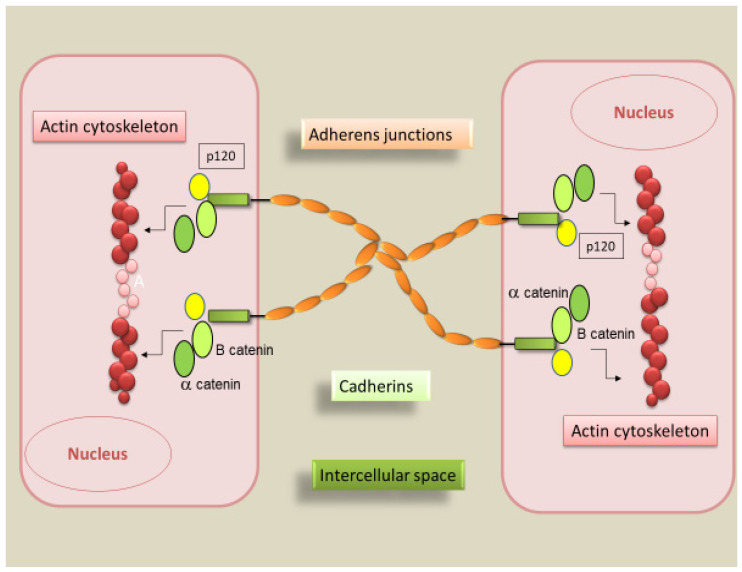
Schematic overview of the molecular components of adhesive intercellular junctions. In adherens junctions, the cadherin cytoplasmic domain binds to p120-α-catenin and β-catenin. The structure, composed of β-catenin, allows α-catenin to link this complex to the cellular actin cytoskeleton.

**Figure 2 ijms-22-13358-f002:**
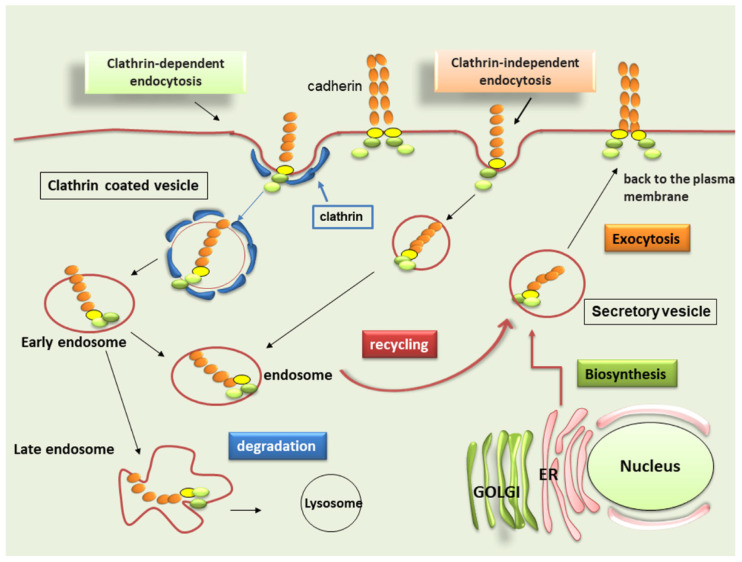
Classic pathways of cadherin trafficking. Cadherins can undergo either clathrin-dependent or independent endocytosis. Trafficking of internalized cadherins occurs through several endosomal compartments. Numerous molecules and endocytic machinery determine the fate of cadherins. After internalization, cadherins are transferred to early endosomes, from which they can be recycled back to the plasma membrane or delivered to late endosomes and lysosomes for degradation.

**Figure 3 ijms-22-13358-f003:**
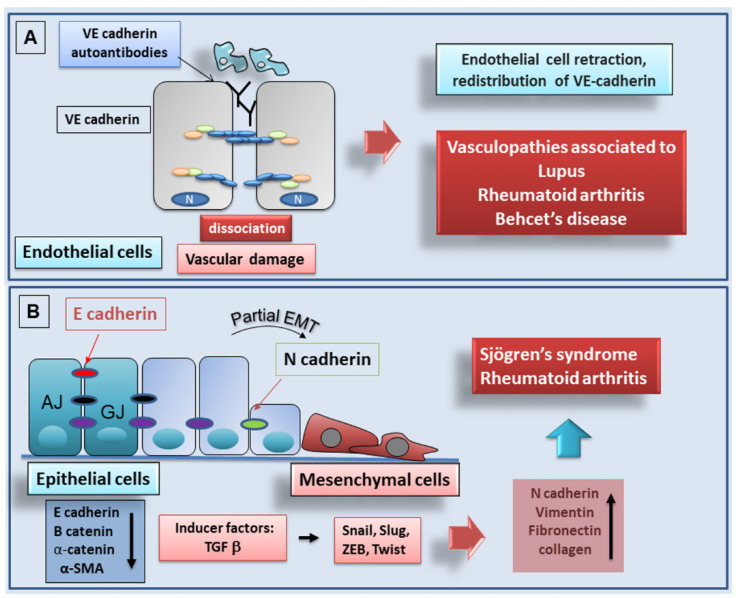
(**A**): potential pathogenic effect of anti-VE cadherin autoantibodies produced in the course of autoimmune diseases. Anti-VE cadherin autoantibodies, by destabilizing endothelial cell-cell interactions, induce increased vascular permeability thereby facilitating leukocyte recruitment and vascular damage. (**B**): schematic representation of the EMT process in rheumatoid arthritis and Sjögren’s syndrome. (AJ, adherens junction; GJ, gap junction).

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
