# Peer review of "Cadherin Signaling in Cancer and Autoimmune Diseases"

_ijms, 2021, doi:10.3390/ijms222413358_

Round 1

Reviewer 1 Report

This paper by Sisto et al is a comprehensive review regarding cadherins and their potential role in cancer and autoimmunity. The paper is mostly well-written, and I find it interesting and of relevance to the research community. The references are both from the 19th century and up to date; maybe I could wish for more new once....

The main overall comment I have is that it is a complex review to read because of the existence of many cadherins and the inclusion of several diseases. I do think that abbreviations could have been avoided some places to make it easier to read. I would also like some of the figures "to stand together", and not as separate figures as it is now. It complicates the reading-flow. I would also like to see a "sum-up" figure regarding how cadherins could be involved in autoimmunity.  

Please also write names of genes in Italics, as is the standard way of annotating genes.

Please also see through whole article and simply sentences where you can!

Minor comments:

*Please rephrase the sentence with "probably" in the abstract.

*Lack of words in line 28

*Write out EMT line 37

*Rephrase in line 44

*Defnine the cadherins before you list them, about line 50

*Line 350: "have been" instead?

*Line 385: rephrase. Turn the sentence around, please

*Line 501: start sentence in another way

Author Response

Manuscript ID: ijms-1479067

Title: Cadherins signalling in cancer and autoimmune diseases

Authors: Margherita Sisto, Domenico Ribatti, Sabrina Lisi

We would like to express our sincere gratitude to the reviewer for his constructive and positive comments and for the very thoughtful critique of our manuscript and are pleased to say that we tried to address all the concerns raised. All changes to the manuscript are highlighted in the text. We respond below in detail to each of the reviewer’s comments and we hope that the reviewer will find satisfactory our responses to his comments.

Major comments:

  1. The references are both from the 19th century and up to date; maybe I could wish for more new once. Where possible, we have replaced older references with recent ones.
  2. The main overall comment I have is that it is a complex review to read because of the existence of many cadherins and the inclusion of several diseases. I do think that abbreviations could have been avoided some places to make it easier to read. We agree with the reviewer that the review is very long and complex due to the huge amount of data reported. However, we believe that the review can enrich the scientific world by playing a supporting role in identifying and clarifying the very varied mechanisms in which cadherins are involved. As suggested, in dealing with the more complex topics we have avoided abbreviations to make reading easier. Furthermore, we remove the paragraph “2.4. Regulation of Cadherins Endocytosis by Growth Factors” because it is too long and does not add any essential information to the review .
  3. I would also like some of the figures "to stand together", and not as separate figures as it is now. It complicates the reading-flow. I would also like to see a "sum-up" figure regarding how cadherins could be involved in autoimmunity. As suggested by reviewer 1 and in accordance with the objections of the other reviewers, we have organized the figures more clearly and brought together some figures. For this reason, updated figures have been included.
  4. Please also write names of genes in Italics, as is the standard way of annotating genes. We made these corrections.
  5. Please also see through whole article and simply sentences where you can! In accordance with the suggestions of all the reviewers we have simplified several manuscript sentences where possible.

Minor comments:

  1. Please rephrase the sentence with "probably" in the abstract.
  2. Lack of words in line 28
  3. Write out EMT line 37
  4. Rephrase in line 44
  5. Define the cadherins before you list them, about line 50
  6. Line 350: "have been" instead?
  7. Line 385: rephrase. Turn the sentence around, please
  8. Line 501: start sentence in another way

All these minor changes were made to the manuscript.

Reviewer 2 Report

The review article by Sisto et al. summarizes the current knowledge of cadherins and their involvement in diseases including, but not limited to cancers. This is a well timed review that may be of broad interest. There are some issues that, I believe, should be addressed.

Overall, there is a lot of language sloppiness all around. This needs to be edited. For example, the very first line in Abstract (line 10) contains a grammar typo. Line 28 has a word “than” missing. Line 42 has a lot of redundancy that does not make sense in current form (“proliferate or not” and then “decide whether to continue or stop proliferating”. Also, “simply die” is not a simple cellular decision as anyone in the field of apoptosis/necrosis will attest to. The sentence in Lines 44-46 makes little grammar sense. The entire paragraph is missing periods at the ends of sentences (this is true throughout the text). Line 121: what is “the meeting point”? Line 124 implies that Macropinocutosis of cadherin comes from Paterson (?). Perhaps it was described by Paterson, but not occurs in him exclusively? “Evidences” should be replaced with “Evidence” in singular.  These are just examples. The entire text should be double checked with careful editing.

The actual review that is appropriate to the title appears to start at Section 4. The first part of the review, section 1-3, appear to be an attempt to write a textbook outside of the scope of a review. These sections are especially protracted. They can be easily removed and replaced with one introductory paragraph, as a suggestion.

Section 3 is titled Role of Cadherins in EMT etc, but the section does not describe their roles. Instead, it describes EMT and changes in cadherin expression, but not roles in cadherins in EMT.

The review overall is very long and for this reason hard to follow. Each section can be easily shortened, including sections 4 on, which will only improve the messages.

Section 4 Line 350. Authors should not be dismissive of metastatic cancer as “couple of diseases”. Cancer and cancer metastasis represent much more than a “couple” of diseases, and metastatic cancer is not just one condition as the authors put it.

Line 363 does not make sense and is plainly wrong: most cancer cells are not known to divert (?) EMT, and most cancers may not involve EMT at all.

Line 364. What is “environmental hint”?

Line 369 and 370 make no sense. “Both” implies two, and at least 3 different conventional therapies are mentioned.

Line 433 – what is “leakages” of E-cadherin, etc?

555-558 – the sentence lists a mishmash of items that do not seem to belong together, plus there is language inconsistency, too.

Line 638: did authors mean RA progression instead of “evolution”?

Expression “very interestingly” should be deleted (Line 681)

Lines 716-720: authors propose “very innovative discoveries” (?), but cite their own review articles. This point may qualify as inappropriate self-citation.

Figures have pretty background, but they appear to carry no message. For example, Figure 3 is supposed to refer to clathrin dependent and independent intake, but only dependent intake is shown. The arrow on the right that points upwards signifies something, but what it is is unclear. Figure 1 looks fine, but it can be easily replaced with a table listing Type 1 and 2 cadherins. The rest has no obvious information. Figure 5 seems to have no point: the figure in its current state can be replaced with 1-2 sentences and clarity will be much improved.

The above points apply in general to most if not all figures. The important parts should be highlighted to drive the point of each figure.

Prof Ribatti is listed as critically reading the review for authors’ contributions. Critical reading of a review by itself is not a valid reason for co-authorship.

Prof Sisto and Listi roles needs to be rephrased – this is a review article that does not generate any data (762-763), so they could not have taken responsibility for the data that were not generated for this review manuscript.

Author Response

Manuscript ID: ijms-1479067

Title: Cadherins signalling in cancer and autoimmune diseases

Authors: Margherita Sisto, Domenico Ribatti, Sabrina Lisi

We would like to express our sincere gratitude to the reviewer for his constructive and positive comments and for the very thoughtful critique of our manuscript and are pleased to say that we tried to address all the concerns raised. All changes to the manuscript are highlighted in the text. We respond below in detail to each of the reviewer’s comments and we hope that the reviewer will find satisfactory our responses to his comments.

Major comments

  1. Overall, there is a lot of language sloppiness all around. This needs to be edited. For example, the very first line in Abstract (line 10) contains a grammar typo. Line 28 has a word “than” missing. Line 42 has a lot of redundancy that does not make sense in current form (“proliferate or not” and then “decide whether to continue or stop proliferating”. Also, “simply die” is not a simple cellular decision as anyone in the field of apoptosis/necrosis will attest to. The sentence in Lines 44-46 makes little grammar sense. The entire paragraph is missing periods at the ends of sentences (this is true throughout the text). Line 121: what is “the meeting point”? Line 124 implies that Macropinocutosis of cadherin comes from Paterson (?). Perhaps it was described by Paterson, but not occurs in him exclusively? “Evidences” should be replaced with “Evidence” in singular.  These are just examples. The entire text should be double checked with careful editing.

We apologize for the errors found in the manuscript and, as suggested, it has been checked.

  1. The actual review that is appropriate to the title appears to start at Section 4. The first part of the review, section 1-3, appear to be an attempt to write a textbook outside of the scope of a review. These sections are especially protracted. They can be easily removed and replaced with one introductory paragraph, as a suggestion.

We agree with the reviewer that the review is very long and complex due to the huge amount of data reported. However, we believe that the review can enrich the scientific world by playing a supporting role in identifying and clarifying the very varied mechanisms in which cadherins are involved. In addition, other reviewers consider the initial part appropriate. However, as suggested, we have tried to shorten some parts by eliminating redundant sentences. Furthermore, we remove the paragraph “2.4. Regulation of Cadherins Endocytosis by Growth Factors” because it is too long and does not add any essential information to the review .

  1. Section 3 is titled Role of Cadherins in EMT etc, but the section does not describe their roles. Instead, it describes EMT and changes in cadherin expression, but not roles in cadherins in EMT.

We agree with the reviewer that the section 3 title does not match the content and we have changed the title appropriately.

  1. Section 4 Line 350. Authors should not be dismissive of metastatic cancer as “couple of diseases”. Cancer and cancer metastasis represent much more than a “couple” of diseases, and metastatic cancer is not just one condition as the authors put it.

We thank the reviewer for this observation and we rewrote the sentence in accordance with his observations.

  1. Line 363 does not make sense and is plainly wrong: most cancer cells are not known to divert (?) EMT, and most cancers may not involve EMT at all.

Line 364. What is “environmental hint”?

Line 369 and 370 make no sense. “Both” implies two, and at least 3 different conventional therapies are mentioned.

The sentence was modified accordingly to the reviewer’s suggestion.

  1. Line 433 – what is “leakages” of E-cadherin, etc?

The sentence was modified.

  1. 555-558 – the sentence lists a mishmash of items that do not seem to belong together, plus there is language inconsistency, too.

The sentence was deleted because it did not improve the meaning of the text.

  1. Line 638: did authors mean RA progression instead of “evolution”?

“progression” is certainly a more appropriate term.

  1. Expression “very interestingly” should be deleted (Line 681)

We delete it, as suggested.

  1. Lines 716-720: authors propose “very innovative discoveries” (?), but cite their own review articles. This point may qualify as inappropriate self-citation.

The sentence was modified accordingly to the reviewer’s suggestion.

  1. Figures have pretty background, but they appear to carry no message. For example, Figure 3 is supposed to refer to clathrin dependent and independent intake, but only dependent intake is shown. The arrow on the right that points upwards signifies something, but what it is is unclear.

The figure 3 was modifies accordingly to the reviewer’s suggestions.

  1. Figure 1 looks fine, but it can be easily replaced with a table listing Type 1 and 2 cadherins.

We preferred the figure to the table because we believe it is easy to interpret.

  1. Figure 5 seems to have no point: the figure in its current state can be replaced with 1-2 sentences and clarity will be much improved.

The above points apply in general to most if not all figures. The important parts should be highlighted to drive the point of each figure.

The figures have been redone and reduced in number in accordance with the correct observations of the reviewer.

  1. Prof Ribatti is listed as critically reading the review for authors’ contributions. Critical reading of a review by itself is not a valid reason for co-authorship.

Prof Sisto and Listi roles needs to be rephrased – this is a review article that does not generate any data (762-763), so they could not have taken responsibility for the data that were not generated for this review manuscript.

We modified  the “Authors Contributions” paragraph, pointing out that all authors contributed to the writing of the manuscript.

Round 2

Reviewer 2 Report

The revised manuscript by Sisto et al addresses most of my points. It needs moderate English editing throughout despite improvements. There are a number of grammar errors, random unnecessary commas inserted seemingly at random, some words missing (one example: line 153), etc, throughout the text. The very first sentence in the abstract (line 10) is grammatically incorrect, and it continues from there. Just one of the numerous examples: line 42 instead of demonstrating it should be demonstrate OR demonstrated. Line 52 should be whose structure (without the comma) instead of “, which”. Again, the same continues throughout the text. Authors should proofread the entirety of the text carefully.

The text is long and winding, with gigantic paragraphs that are very hard to follow. Figures, while improved, do not add much to the text, really, with Figure 1 being the worst in terms of carrying a message. Why dedicate an entire figure to two classes of cadherins when the two classes, according to the figure, are identical. What do orange versus blue m&ms signify? Different sequence motifs or absence of HAV motif? In case of the latter, the legend and colors would be completely redundant.

In sum, the text might be of interest to those in the field who could muster the energy to read through its bulk.

Author Response

Manuscript ID: ijms-1479067

Title: Cadherins signalling in cancer and autoimmune diseases

Authors: Margherita Sisto, Domenico Ribatti, Sabrina Lisi

We would like to express our sincere gratitude to the reviewer for his patience. The entire manuscript was reviewed by a professional scientific text editor. Furthermore, as suggested, we have eliminated figure 1 and its legend. We hope to have improved the manuscript in this way.